# Non-linear association of cardiometabolic index with gallstone disease in US adults: A cross-sectional study

Zhe Xiong[1], Qiuyue Sun[1,2], Jin Huang[1], Fengdong Li[1]*

1 Department of Gastroenterology, Changzhou second Peoples Hospital Affiliated to Nanjing Medical University, Changzhou, Jiangsu Province, China, 2 Graduate School of Nanjing Medical University, Nanjing, Jiangsu Province, China

These authors contributed equally to this work.
* fengdongli@njmu.edu.cn

## Abstract

### Objective

Obesity and disorders of lipid metabolism are key factors in gallstone formation. The cardiometabolic index (CMI) is a new marker combining obesity indicators (WHtR) and lipid levels (TG/HDL-C). The aim of this study was to investigate whether CMI is associated with the prevalence of gallstone disease (GSD) in adults.

### Methods

This study was a cross-sectional study by cohort data from the National Health and Nutrition Examination Survey (NHANES) January 2017-March 2020 cycle. Using sample-weighted multivariate logistic regression analysis, sample weighted restricted cubic spline (RCS), the receiver operating characteristic (ROC) and subgroup analysis to assist us with gaining insight into the association between CMI and GSD.

### Results

Among the 2825 included participants, 307 of them were patients with GSD. In a model adjusting for multiple covariates, each unit increase in CMI increased the odds of GSD by 16% (OR = 1.16; 95% CI = 1.04-1.30; $P$ = 0.015). Compared to participants with the lowest CMI, those with the highest index had a significantly increased risk of GSD (OR = 2.52; 95% CI = 1.45-4.36; $P$ = 0.006). The sample-weighted multivariable adjusted RCS revealed a nonlinear relationship between CMI and GSD. ROC analysis revealed that CMI had a high predictive effect on the diagnosis of GSD (AUC = 0.680). Subgroup analysis suggested the influence of gender differences on the relationship between CMI and GSD.

**Data availability statement:** All data used for analysis in the study are available from https://figshare.com, DOI: https://10.6084/m9.figshare.28323725.

**Funding:** This work was supported by the Fund of Changzhou Medical Center, Nanjing Medical University (CMCC202209 and CMCM202310). The funder of this work is Jin Huang, who participated in the Data curation and Formal Analysis of this study.

**Competing interests:** The authors have declared that no competing interests exist.

## Conclusions

The present study found that elevated CMI significantly increased the risk of GSD and emphasized the non-linear relationship between both. These results suggest that CMI could be a potential marker for early screening and intervention of GSD, thus providing guidance for future clinical management of GSD.

## Introduction

Gallstone disease (GSD) as one of the most common digestive disorders, it is present in populations all over the world [1,2]. The prevalence and incidence of GSD have been reported to be approximately 15% and 0.6% per year, respectively. The prevalence of gallstones increases with age, and it is more prevalent among women than among men [3–5]. Gallstones can be located anywhere in the biliary system and are mainly categorized as cholecystolithiasis, calculus of extrahepatic bile duct and hepatolithiasis [1]. Most patients with GSD are asymptomatic, but when gallstones enter the bile ducts and cause obstruction, the patient experiences typical biliary colic, mostly accompanied by nausea and vomiting [4]. Approximately 1% to 4% of symptomatic patients with GSD develop serious complications, such as acute cholecystitis, gallstone pancreatitis, Mirizzi syndrome, and gallstone ileus [6–8]. In addition, long-term stimulation by gallstones and inflammation may increase the risk of gallbladder cancer incidence [9,10]. In the U.S., more than 800,000 cholecystectomies are performed each year, costing nearly $6.5 billion annually, making GSD a major global health concern [11]. Therefore, it is crucial to find accurate and reliable clinical indicators in order to provide new insights for early screening and intervention of GSD.

According to previous studies, risk factors for GSD include age, female gender, pregnancy, unhealthy lifestyle, and inflammatory diet [12–14]. one of the key factors contributing to the formation of gallstones is obesity [5]. It is always known that visceral fat plays a supporting, stabilizing and protective role for the internal organs of the body [15]. However, excessive accumulation of visceral fat can lead to metabolic disorders related to obesity, such as insulin resistance, glucose intolerance and dyslipidemia, which undoubtedly increase the risk of gallstone disease [16–18]. Multiple studies have clearly demonstrated that obesity significantly increases the risk of gallstones and have emphasized the important role of novel anthropometric indices in the prevention of gallstones [19,20]. In addition, disorders of lipid metabolism in the body also promote the gallstones formation. Triglyceride (TG), as the lipid component of blood lipids, is an important product of lipid metabolism, and an increase in serum TG can induce a variety of metabolism-related diseases. However, high-density lipoprotein cholesterol (HDL-C) transports cholesterol from surrounding tissues, which is then converted to bile acids or excreted directly from the intestine via bile [21]. Normally, cholesterol, lecithin, and bile salts in bile work together to stabilize the bile. As TG increases or HDL decreases, cholesterol becomes supersaturated, causing cholesterol crystals to precipitate to form stones [22]. Studies have shown that serum HDL levels are inversely and linearly related to risk of gallstone. However, the risk of gallstones increases with

increasing serum TG levels [23]. Although many studies have suggested a strong association between obesity and lipid metabolism and GSD, there is still a lack of reliable clinical indicators to manage the occurrence of gallstones.

The cardiometabolic Index (CMI) is a new index calculated from waist-to-height ratio (WHtR), triglycerides, and high-density lipoprotein cholesterol [24]. In contrast to previous body roundness index (BRI) and lipid accumulation product (LAP) indicators, CMI provides a more complete picture of the body's metabolic state through an individual's degree of obesity and lipid levels and has been found to have better predictive potential in some studies [25–27]. As the study evolved, researchers found that CMI could not only assess heart health status it could also be used to predict the presence and severity of metabolic syndrome in adult obese patients [28]. Furthermore, elevated CMI has been demonstrated to be significantly associated with insulin resistance and type 2 diabetes [29]. Based on the above findings, it is considered that metabolic syndrome (MetS) and insulin resistance are thought to be associated with an increased risk of gallstones [30,31]. Therefore, the present study hypothesized that metabolic abnormalities reflected by changes in CMI may influence the occurrence of gallstones. This study, with data from the National Health and Nutrition Examination Survey (NHANES), illustrates the relationship between CMI and GSD in U.S. adults for the first time, ultimately providing additional guidance on the clinical management of GSD and offering important insights for future research.

## Methods

### Study population and research design

The NHANES program was committed to collecting comprehensive data regarding the health and nutrition status of the American population, as well as relevant health behavior information. It covered demographics, dietary, examination, laboratory, and questionnaire data. Visit http://www.cdc.gov/nhanes to access more information about NHANES.

There were 15,560 participants enrolled in this study between January 2017 and March 2020, of whom 9232 were adults aged ≥ 20. However, 5,483 participants lacking CMI data and 6 participants lacking GSD data were excluded. Moreover, 918 participants were excluded because they did not have data on education status, poverty to income ratio (PIR), body mass index (BMI), smoking, alcohol consumption, hypertension, diabetes, cancer, and coronary heart disease (CHD). **Fig 1** illustrated the flowchart of the screening process of the present study.

### Cardiometabolic index

Reference to previously published study [24], information on fasting TG and HDL in CMI was obtained from laboratory data, and information on waist circumference (WC) and height was obtained from measurement data. The specific calculations are as follows:

$$WHtR = \text{waist circumference (cm)/height (cm)}$$

$$CMI = \text{TG (mmol/L)/HDL-C (mmol/L)} \times WHtR$$

### Assessment of gallstone

Questionnaires from the NHANES data were used for the diagnosis of gallstones. The questionnaire, "Have you been told by a doctor or other health professional that you have gallstones?" was used to indicate the presence of gallstones when the participant answered "yes".

### Covariables

Several covariates were also included in this study for statistical analysis, as they may be potential confounding factors affecting the relationship between CMI and GSD. Confounding factors that were included age, gender, race, education

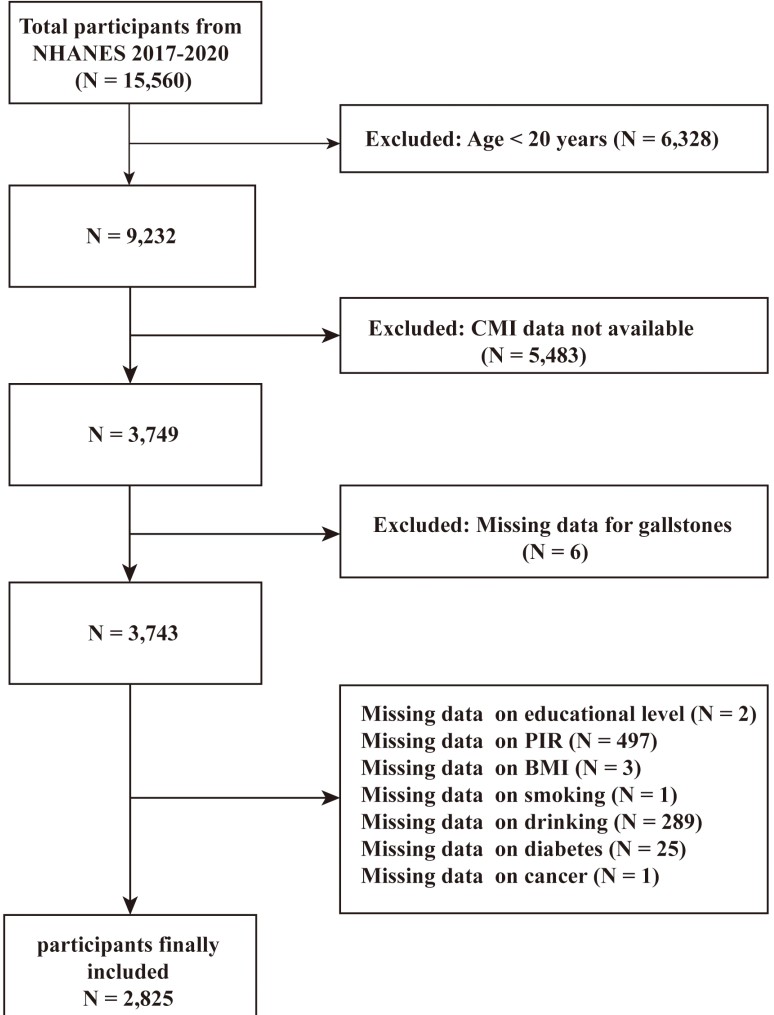

**Fig 1. The flowchart of participants selection.** NHANES: National Health and Nutrition Examination Survey; CMI: cardiometabolic index.

level, PIR, and BMI. Smoking more than 100 cigarettes during one's lifetime was identified as smoking status. Ever have 4/5 or more drinks every day was identified as drinking status. The results for total cholesterol (TC) were obtained from laboratory data. Definition of physical activity, hypertension, diabetes, cancer, and CHD were obtained from question-naires or clinical diagnosis.

## Statistical analysis

To ensure the reliability of the population estimates, this study processed the data according to sample weight recom-mended by NHANES. In the analysis of continuous variables, the researchers used the weighted mean and standard deviation (SD), and in the analysis of categorical variables, they used the weighted percentages. CMI was grouped into quartiles based on the characteristics of continuous variables. The Kruskal-Wallis H-test or chi-square test was applied to compare the differences between quartile groupings of CMI. Weighted multivariate logistic regression models were constructed to evaluate the relationship between CMI and GSD using the odds ratio (OR) and 95% confidence interval (CI). Logistic regression is a generalized linear model used for addressing classification problems, and it can intuitively

reflect the impact of changes in CMI on the risk of GSD. The crude model (Model 1) was not adjusted. Age, sex, and race were accounted for in Model 2. Based on Model 2, Model 3 had adjustments for education level, PIR, physical activity, smoking status, drinking status, TC, hypertension, diabetes mellitus, cancer, and CHD. The adjustment did not account for BMI, as it had a substantial impact on exposure. Additionally, we further analyzed the nonlinear relationship between CMI and the prevalence of GSD using restricted cubic splines (RCS), generalized additive models (GAM), and smooth curve fitting. RCS is a method used for modeling nonlinear relationships; it segments the values of the independent variable and employs cubic spline functions within each interval for modeling. Therefore, RCS can effectively capture the complex relationship between CMI and GSD. Subsequently, the receiver operating characteristic (ROC) curve was used to calculate the area under curve (AUC) value to assess the diagnostic value of the CMI index for GSD. Finally, subgroup analyses were conducted to test for interactions between CMI and specific covariates. Statistical significance was determined at a threshold of $P < 0.05$ during the statistical analysis. The statistical analyses conducted in this study were conducted using R version 4.3.1.

## Results

### Baseline characteristics

After screening, the analysis encompassed a total of 2,825 participants. Table 1 presents the baseline characteristics of the study population by CMI quartiles. As indicated in the table, there were significant differences in CMI quartiles by age, gender, race, education level, PIR, BMI, hypertension, diabetes, and TC ($P < 0.05$). The increase in CMI was associated with an increase in the proportion of Mexican Americans, an increase in the proportion less than high school, and a decrease in annual household income. It is noteworthy that in the highest quartile there is a higher proportion of males than females (62.16% vs. 37.8%), with the proportion of males showing an upward trend in the quartiles, while the opposite is true for females. As CMI increased, BMI of the participants also increased. In addition, the prevalence of hypertension and diabetes also increased with increasing CMI. A significant correlation between CMI quartiles and gallstone prevalence without adjustment for covariates ($P = 0.030$).

### Association between the CMI index and GSD

To provide insight into the potential link between CMI and GSD, the sample-weighted multivariate logistic regression models were constructed (Fig 2) (S1 Table). As shown in this table, when CMI was analyzed as a continuous variable, the prevalence of GSD was significantly correlated with CMI. (OR = 1.16; 95% CI = 1.04–1.30; $P = 0.015$) under model 3, which considered all relevant covariates. This suggested that for every unit increase in CMI, the odds of GSD increased by 16%. Next, CMI was further categorized into quartiles to describe the correlation. In model 3, Participants in the highest quartile (Q4) of CMI increased the risk of developing GSD by 152% (OR = 2.52; 95% CI = 1.45–4.36; $P = 0.006$) compared to those in the lowest quartile (Q1), and this positive correlation remains significant in both Model 1 and Model 2 species. Furthermore, the results of trend analysis for all three models were statistically significant ($P$ for trend < 0.05). This indicated that as CMI increases, the prevalence of GSD is higher. Finally, the sample-weighted multivariable adjusted RCS revealed a statistically significant nonlinear relationship between CMI and GSD ($P$ nonlinear = 0.0121) (Fig 3). Additionally, generalized additive models and smooth curve fitting showed a nonlinear relationship between CMI and OAB (log-likelihood ratio test P value < 0.001) (Table 2).

### Diagnostic value of CMI index for GSD

ROC curves were further plotted to assess the diagnostic value of CMI and its components for GSD. As shown in Fig 4 in diagnosing GSD, CMI has the highest accuracy with an AUC of 0.680. The other indices with diagnostic values in descending order were WHtR (AUC = 0.646) and TG/HDL (AUC = 0.570).

**Table 1. The baseline characteristics of the study population by CMI quartiles in the logistic regression model.**

| Characteristic | Total | Quartiles of Cardiometabolic index (CMI) | | | | P value |
|---|---|---|---|---|---|---|
| | (N=2825) | Q1 (N=707) 0.027-0.255 | Q2 (N=706) 0.255-0.459 | Q3 (N=704) 0.459-0.816 | Q4 (N=708) > 0.816 | |
| Age (years) | 47.72±0.76 | 43.03±1.40 | 48.93±0.89 | 49.99±1.21 | 49.38±0.94 | < 0.001 |
| Gender (%) | | | | | | < 0.001 |
| Male | 1449 (50.13%) | 306 (41.63%) | 347 (46.93%) | 365 (49.77%) | 431 (62.16%) | |
| Female | 1376 (49.87%) | 401 (58.37%) | 359 (53.07%) | 339 (50.23%) | 277 (37.84%) | |
| Race (%) | | | | | | < 0.001 |
| Mexican American | 359 (8.67%) | 57 (6.17%) | 82 (7.71%) | 86 (9.32%) | 134 (11.58%) | |
| Other Hispanic | 269 (6.27%) | 50 (4.77%) | 69 (7.29%) | 72 (7.28%) | 78 (5.97%) | |
| Non-Hispanic White | 1069 (66.02%) | 254 (67.87%) | 250 (65.89%) | 250 (59.91%) | 315 (69.52%) | |
| Non-Hispanic Black | 695 (10.18%) | 232 (13.36%) | 204 (11.63%) | 181 (11.73%) | 78 (4.20%) | |
| Other Race | 433 (8.86%) | 114 (7.83%) | 101 (7.48%) | 115 (11.76%) | 103 (8.73) | |
| Education level (%) | | | | | | 0.004 |
| Less than high school | 459 (9.24%) | 71 (5.56%) | 96 (7.84%) | 129 (10.94%) | 163 (12.87%) | |
| High school or GED | 673 (25.22%) | 163 (21.66%) | 169 (24.50%) | 171 (26.61%) | 170 (28.35%) | |
| Above high school | 1693 (65.54%) | 473 (72.78%) | 441 (67.66%) | 404 (62.45%) | 375 (58.78%) | |
| PIR | 3.20±0.07 | 3.45±0.10 | 3.23±0.12 | 3.10±0.11 | 3.00±0.10 | 0.008 |
| BMI (kg/m²) | 29.78±0.20 | 24.94±0.21 | 28.91±0.40 | 31.55±0.38 | 34.04±0.38 | < 0.001 |
| Alcohol (%) | | | | | | 0.189 |
| Yes | 442 (14.13%) | 98 (11.16%) | 114 (14.84%) | 105 (13.21%) | 125 (17.28%) | |
| No | 2383 (85.87%) | 609 (88.84%) | 592 (85.16%) | 599 (86.79%) | 583 (82.72%) | |
| Smoked (%) | | | | | | 0.274 |
| Yes | 564 (17.31%) | 127 (13.69%) | 149 (17.93%) | 135 (19.38%) | 153 (18.62%) | |
| No | 2261 (82.69%) | 580 (86.31%) | 557 (82.07%) | 569 (80.62%) | 555 (81.38%) | |
| Physical activity (%) | | | | | | < 0.001 |
| Yes | 1394 (58.11%) | 418 (69.54%) | 360 (57.79%) | 317 (55.74%) | 299 (48.82%) | |
| No | 1431 (41.89%) | 289 (30.46%) | 346 (42.21%) | 387 (44.26%) | 409 (51.18%) | |
| Hypertension (%) | | | | | | < 0.001 |
| Yes | 1277 (38.13%) | 220 (22.33%) | 304 (36.16%) | 371 (46.14%) | 382 (49.21%) | |
| No | 1548 (61.87%) | 487 (77.67%) | 402 (63.84%) | 333 (53.86%) | 326 (50.79%) | |
| Diabetes (%) | | | | | | < 0.001 |
| Yes | 491 (11.92%) | 40 (2.53%) | 101 (8.21%) | 142 (13.15%) | 208 (23.96%) | |
| No | 2334 (88.08%) | 667 (97.47%) | 605 (91.79%) | 562 (86.85%) | 500 (76.04%) | |
| CHD (%) | | | | | | 0.053 |
| Yes | 130 (3.68%) | 14 (2.03%) | 32 (4.04%) | 35 (3.41%) | 49 (5.28%) | |
| No | 2695 (96.12%) | 693 (97.97%) | 674 (95.96%) | 669 (96.59%) | 659 (94.72%) | |
| Cancers (%) | | | | | | 0.207 |
| Yes | 307 (11.76%) | 68 (10.33%) | 84 (13.50%) | 64 (9.47%) | 91 (13.51%) | |
| No | 2518 (88.24%) | 639 (89.67%) | 622 (86.50%) | 640 (90.53%) | 617 (86.49%) | |
| Gallstones (%) | | | | | | 0.030 |
| Yes | 307 (11.69%) | 35 (6.45%) | 76 (12.23%) | 93 (12.95%) | 103 (15.44%) | |
| No | 2518 (88.31%) | 672 (93.55%) | 630 (87.77%) | 611 (87.05%) | 605 (84.56%) | |
| WC (cm) | 100.79±0.49 | 87.43±0.467 | 99.35±0.85 | 105.17±0.82 | 112.00±1.04 | < 0.001 |
| TC (mmol/L) | 4.79±0.05 | 4.53±0.07 | 4.78±0.07 | 4.88±0.06 | 4.99±0.08 | < 0.001 |
| TG (mmol/L) | 1.26±0.03 | 0.56±0.01 | 0.89±0.014 | 1.26±0.01 | 2.32±0.08 | < 0.001 |
| HDL (mmol/L) | 1.39±0.02 | 1.77±0.03 | 1.46±0.02 | 1.27±0.01 | 1.05±0.01 | < 0.001 |

Mean±SD for continuous variables, the P value was calculated by weighted linear regression; frequency percentages for categorical variables, the P value was calculated by weighted chi-square test. PIR: poverty-to-income ratio; BMI: Body Mass Index; CHD: coronary heart disease; CMI: cardiometabolic index; WC: Waist circumference; TG: Triglycerides; TC: Total cholesterol; HDL: High-density lipoprotein cholesterol.

| Characteristic | Odd Ratios (95%CI) | p value | p for trend |
|---|---|---|---|
| **CMI** | | | |
| Model 1 | 1.13 (1.01, 1.27) | 0.041 | |
| Model 2 | 1.22 (1.05, 1.41) | 0.012 | |
| Model 3 | 1.16 (1.04, 1.30) | 0.015 | |
| **Quartile of CMI** | | | |
| **Model 1** | | | **< 0.001** |
| Q1 | Reference | | |
| Q2 | 2.02 (1.05, 3.88) | 0.036 | |
| Q3 | 2.16 (0.92, 5.06) | 0.074 | |
| Q4 | 2.65 (1.68, 4.19) | < 0.001 | |
| **Model 2** | | | **< 0.001** |
| Q1 | Reference | | |
| Q2 | 1.96 (0.98, 3.90) | 0.056 | |
| Q3 | 2.09 (0.89, 4.94) | 0.087 | |
| Q4 | 3.12 (1.92, 5.05) | < 0.001 | |
| **Model 3** | | | **0.008** |
| Q1 | Reference | | |
| Q2 | 1.81 (0.78, 4.19) | 0.129 | |
| Q3 | 1.87 (0.66, 5.29) | 0.181 | |
| Q4 | 2.56 (1.42, 4.60) | 0.009 | |

**Fig 2. Association between CMI and gallstone disease (GSD) in NHANES 2017-2020, weighted.** Model 1: unadjusted. Model 2: age, sex and race were adjusted. Model 3: age, sex, race, education level, PIR, smoking status, drinking status, physical activity, total cholesterol, hypertension, diabetes, cancer, and coronary heart disease were adjusted. CMI: cardiometabolic index; OR: odds ratio; 95% CI: 95% confidence interval.

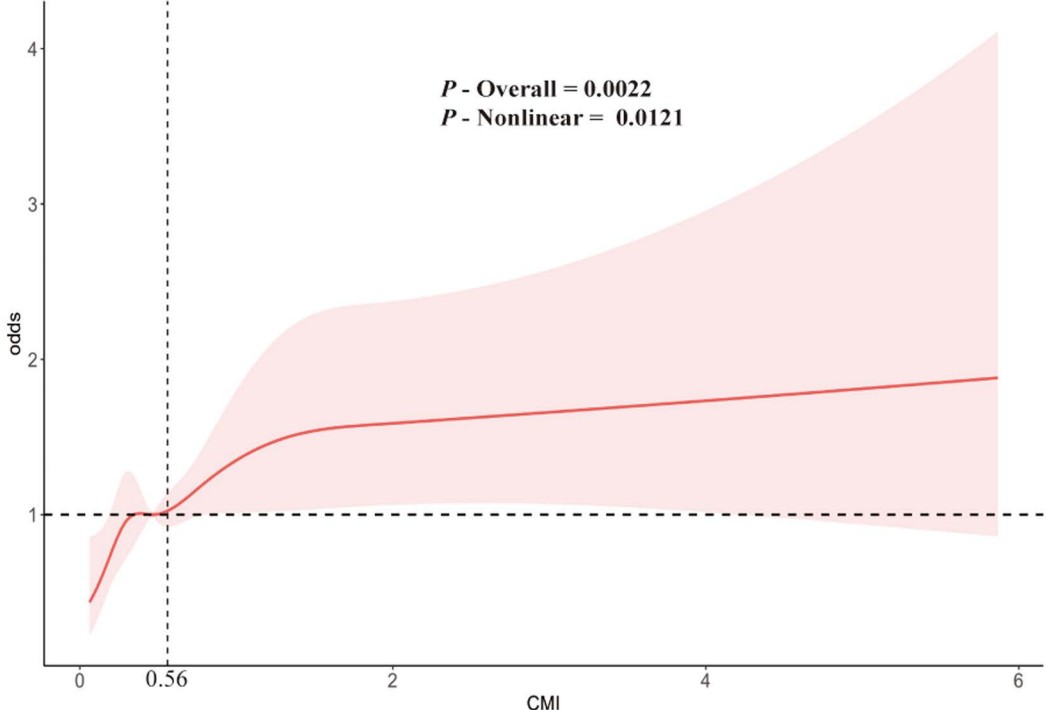

**Fig 3. The restricted cubic spline for the association between CMI and gallstone.**

**Table 2. Threshold effect analysis of the association between CMI and female gallstone disease (GSD) prevalence.**

| Outcome: GSD | Adjusted OR (95%CI) | P value |
|---|---|---|
| Fitting by standard linear model | 0.013 (0.001, 0.025) | 0.036 |
| Fitting by two-piecewise linear model | | |
| Inflection point | 0.561 | |
| CMI < 0.561 | 0.165 (0.09, 0.239) | <0.001 |
| CMI > 0.561 | 0.003 (−0.01, 0.016) | 0.668 |
| Logarithmic likelihood ratio test P value | | < 0.001 |

The associations were adjusted for age, sex, race, education level, PIR, smoking status, drinking status, physical activity, total cholesterol, hypertension, diabetes, cancer, and coronary heart disease.

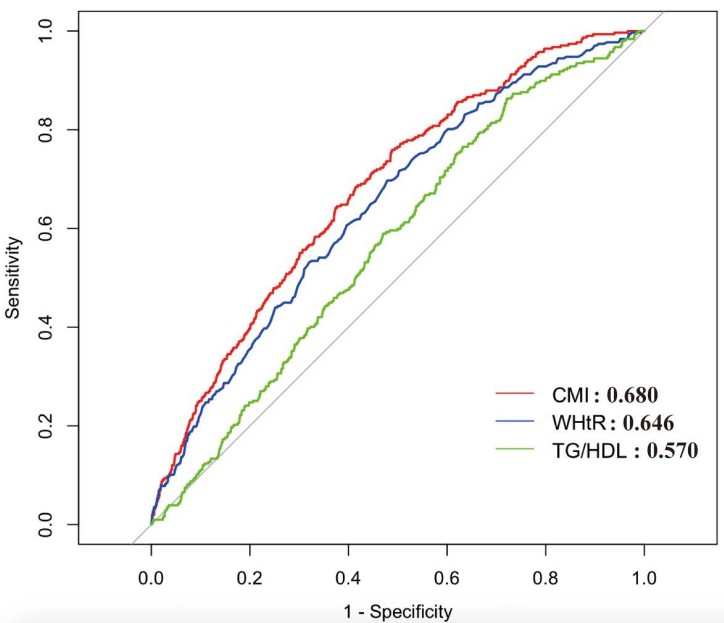

**Fig 4. The ROC curve of CMI, TG/HDL and WHtR for the diagnosis of gallstones.**

## Subgroup analysis

Completely adjusted multivariate logistic regression analyses were used for each subgroup to assess that the association between CMI and GSD was robust across age, sex, education level, smoking, alcohol consumption, hypertension, diabetes, cancer, and CHD stratification. As shown in S2 Table **and Fig 5**, the positive correlation between CMI and gallstones was robust in all subgroups (*P* for interaction > 0.05) except in the gender subgroup (*P* for interaction = 0.015).

## Discussion

Through cross-sectional analysis of NHANES data, we discovered that the prevalence of GSD was substantially and positively correlated with CMI. In the fully adjusted model, the risk of GSD increased by 152% for each unit increase in CMI, as indicated by the results of this study. In all three models, A trend test produced significant significance, indicating a

| Subgroup | Odd Ratios (95%CI) | p value | | p for interaction |
|---|---|---|---|---|
| **Age (year)** | | | | **0.132** |
| < 50 | 1.12 (0.97, 1.28) | 0.105 | | |
| ≥ 50 | 1.22 (1.05, 1.42) | 0.017 | | |
| **Gender** | | | | **0.015** |
| Male | 1.09 (0.88, 1.34) | 0.385 | | |
| Female | 1.57 (1.09, 2.25) | 0.021 | | |
| **Education level** | | | | **0.931** |
| Less than high school | 1.18 (0.93, 1.51) | 0.158 | | |
| High school or GED | 1.21 (0.70, 2.10) | 0.456 | | |
| Above high school | 1.14 (0.99, 1.31) | 0.059 | | |
| **Alcohol** | | | | **0.788** |
| Yes | 1.11 (0.77, 1.60) | 0.525 | | |
| No | 1.16 (1.01, 1.33) | 0.039 | | |
| **Smoked** | | | | **0.494** |
| Yes | 1.52 (1.01, 2.27) | 0.045 | | |
| No | 1.10 (0.96, 1.26) | 0.133 | | |
| **Hypertension** | | | | **0.138** |
| Yes | 1.27 (1.08, 1.48) | 0.008 | | |
| No | 1.06 (0.91, 1.24) | 0.391 | | |
| **Diabetes** | | | | **0.599** |
| Yes | 1.17 (0.97, 1.42) | 0.086 | | |
| No | 1.13 (0.98, 1.31) | 0.079 | | |
| **Cancer** | | | | **0.097** |
| Yes | 1.69 (1.14, 2.50) | 0.015 | | |
| No | 1.13 (1.01, 1.26) | 0.037 | | |
| **Coronary heart disease** | | | | **0.532** |
| Yes | 0.77 (0.21, 2.80) | 0.627 | | |
| No | 1.16 (1.03, 1.31) | 0.021 | | |

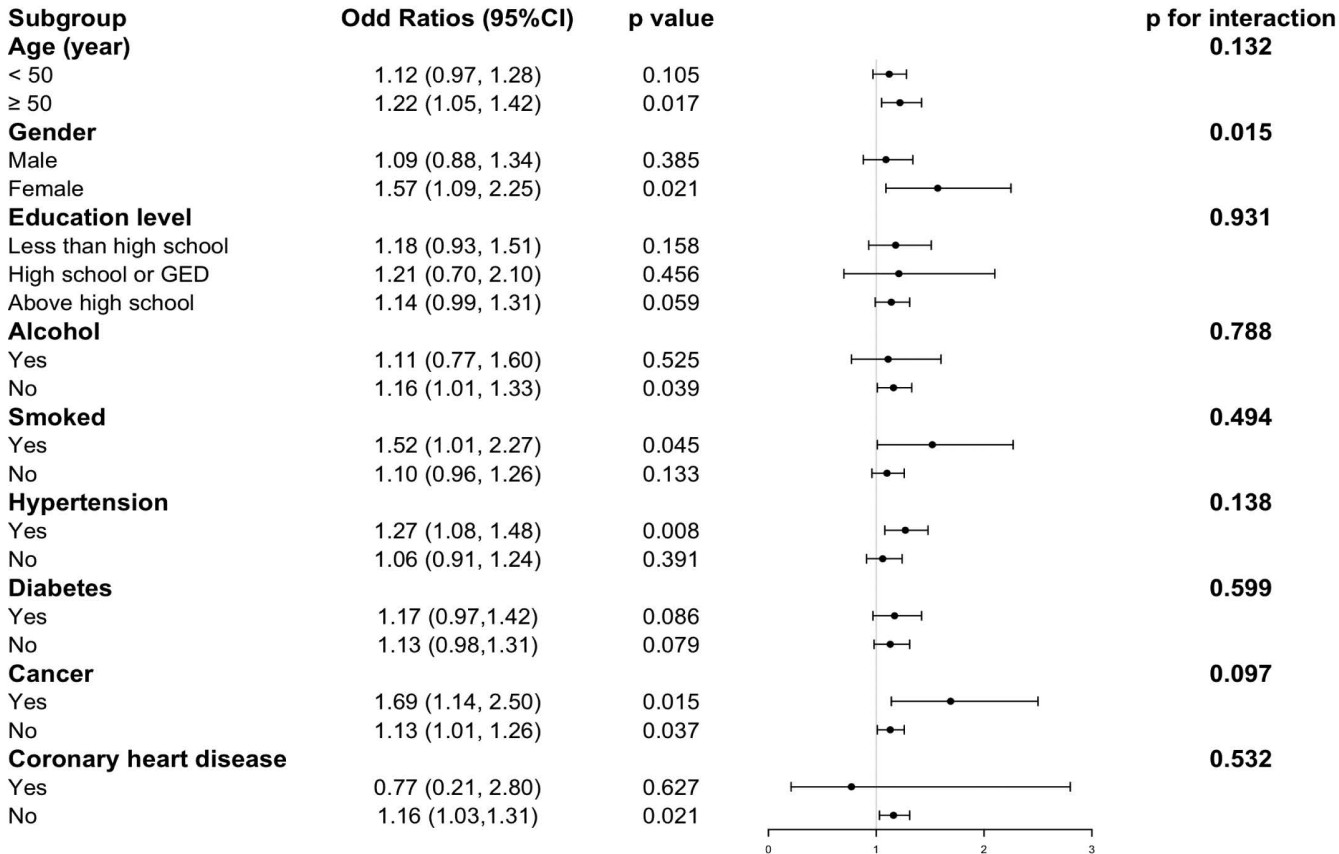

**Fig 5. Subgroup analysis between CMI and gallstone disease (GSD).** Note: All covariates (as in Model 3) were adjusted except the stratification variable itself. CMI: cardiometabolic index; OR: odds ratio; 95% CI: 95% confidence interval.

dose-response relationship between CMI and GSD. Also, the RCS confirmed there is a nonlinear association between the two. Moreover, the ROC analysis highlighted the promising diagnostic value of CMI in the diagnosis of GSD. Subgroup analysis and interaction testing demonstrated that the association between CMI and GSD was steady in age, education level, smoking, alcohol consumption, hypertension, diabetes, cancer, and CHD. In summary, CMI may be regarded as an effective tool for screening high-risk groups for GSD, which is important for early intervention and personalized treatment of GSD.

The CMI, as an indicator to assess the degree of obesity and lipid levels in an individual, was thought to provide a better insight into the body's metabolic status. In previous studies, obesity resulted disorders of lipid metabolism and excessive accumulation of cholesterol, which were important mechanisms for gallstone formation [21,32]. As a result, obesity is universally acknowledged as a concrete risk factor for gallstones. Furthermore, a study from a health check-up cohort of Chinese adult emphasized that obesity promoted gallstones development by investigating the association between metabolically health obesity (MHO) and gallstones [12]. Several anthropometric measures reflecting obesity, such as BMI, WC, and WHtR, have been demonstrated to be independently linked to the occurrence of gallstones [33]. Furthermore, the triglyceride glucose-waist height ratio (TyG-WHtR) was superior to triglyceride glucose-body index mass (TyG-BMI) and triglyceride glucose-waist circumference (TyG-WC) in identifying gallstone risk in a new indicator for assessing insulin resistance [34]. Similarly, obesity affects cholesterol metabolism, which is a key part of gallstone formation. Some serum

lipid markers have been shown to be important factors in gallstone risk. Based on a multicenter study and meta-analysis, Zhang et al. found that both low HDL cholesterol levels and high TG levels were risk factors for gallstone [23]. In addition, multivariate Mendelian randomization analysis also revealed that serum TG was an independent risk factor to gallstones, and lowering TG also lower the risk of gallstones [35].

To date, the association of CMI with metabolism-related diseases had been elaborated in numerous literatures. On the one hand, CMI was pointed out to have better predictive potential in identifying MetS in U.S. adults [28]. On the other hand, higher CMI was associated with an increased prevalence of NAFLD and exacerbation of liver fibrosis [36]. However, there were no research that explored the association between CMI and GSD. Therefore, the prospective association between CMI and GSD was first disclosed in this study using NHANES data. The present study, the sample-weighted multivariate logistic regression models suggested a significant positive association between CMI and the risk of developing GSD and demonstrated a nonlinear relationship. These findings are beneficial for identifying asymptomatic GSD patients in clinical practice. Given that waist-to-height ratio and lipid profiling are routine examinations, CMI is far more accessible compared to abdominal ultrasound. Therefore, CMI can serve as a novel and convenient indicator for GSD prevalence and may assist clinicians in determining whether patients should undergo abdominal ultrasound screening in the future. Additionally, interaction tests revealed that the association between CMI and gallstones exhibited statistically significant variations across subgroups stratified by age, sex, alcohol consumption, smoking status, hypertension, diabetes mellitus, cancer, and CHD. The findings highlight that a higher CMI is associated with an elevated prevalence of GSD in females. This may be attributed to higher estrogen levels in women, which promote cholesterol deposition and influence gallstone formation. In animal experiments, GPER antagonists prevented estrogen-induced cholesterol stone formation in female mice [37]. Based on the results of an epidemiologic survey, estrogen therapy in menopausal women significantly increases the risk of symptomatic gallstones and cholecystectomy [38]. However, the clinical relevance of identifying emergent GSD using the CMI is extended by acknowledging the significant relationship between GSD and CVD, probably due to common causal pathways, as established through meta-analysis and meta-regression [39]. In conclusion, all these reports further prove the credibility of the findings of this research.

Regarding the potential mechanism between CMI and GSD, we first hypothesized that it could be attributed to inflammation and oxidative stress. Adipose tissue secretes a diverse array of inflammatory factors in addition to its function as an energy storage and organ protection, such as IL-1β, TL-6 and TNF-α[40]. These inflammatory factors can lead to gallstone formation by directly affecting the contractile function of gallbladder epithelial cells [41]. Furthermore, reactive oxygen species (ROS) produced during the inflammatory process mediate gallbladder epithelial cell damage, thereby promoting cholecystitis and gallstones [42]. Second, immune cell infiltration may also be the mechanism by which lipid accumulation leads to GSD. Adipose tissue contains a large number of immune cells who are able to regulate adipocyte function by secreting a variety of factors under physiological and pathological conditions [40]. Accumulation of adipose tissue macrophages (ATM) has been reported to affect cholesterol metabolism in obese states [43,44]. Besides, the researchers found that inhibiting the formation of neutrophil extracellular traps (NETs) was effective in suppressing crystal aggregation in bile [45,46]. Third, adipose tissue induced insulin resistance is similarly a key factor for stone formation. In the liver, insulin resistance promotes cholesterol secretion through ABCG5/8 induced aberrant expression of FOXO1 [47]. The above studies provided further strong evidence that CMI has a positive correlation with an elevated risk of GSD in terms of pathological mechanisms.

This investigation boasts numerous noteworthy advantages: first, this research considered the sample-weighted design from the NHANES database, which is nationally representative. Second, covariates that could potentially influence the relationship between CMI and GSD were appropriately adjusted to ensure the accuracy of the results. Third, subgroup analysis of the population was conducted in this study and further validated the stability of the results. However, we accept that the current study is subject to certain inherent limitations. First, the causative relationship between CMI and GSD could not be ascertained since this investigation was predominantly founded on a cross-sectional study. Second, due to

the limitations of the NHANES database, we were unable to include all potential confounding factors that may influence the relationship between CMI and GSD, such as dietary habits and genetic predisposition. Lastly, we must emphasize that the outcome variable and some covariates in this study are based on self-reported data, which may introduce recall bias. Therefore, the findings of this study require validation through longitudinal research.

## Conclusions

In summary, the current investigation revealed that the higher CMI was strongly associated with higher prevalence of GSD and confirmed a nonlinear relationship between the two. Subgroup analyses indicate that the association varies between the genders. These findings may help physicians to screen for people at risk of GSD and highlight the importance of taking gender differences into account when delivering personalized interventions for accurate gallstone prevention.

## Supporting information

**S1 Table. Association between CMI and gallstone disease (GSD) in NHANES 2017–2020, weighted.**
(DOCX)

**S2 Table. Subgroup analysis between CMI and gallstone disease (GSD).**
(DOCX)

## Author contributions

**Conceptualization:** Zhe Xiong, Fengdong Li.

**Data curation:** Jin Huang.

**Formal analysis:** Jin Huang.

**Funding acquisition:** Jin Huang.

**Methodology:** Qiuyue Sun, Fengdong Li.

**Software:** Qiuyue Sun.

**Writing – original draft:** Zhe Xiong.

**Writing – review & editing:** Zhe Xiong, Fengdong Li.

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
