## [Decision Letter · Decision Letter 0]

PONE-D-24-47105Non-linear association of cardiometabolic index with gallstone disease in US adults: a cross-sectional studyPLOS ONE

Dear Dr. Li,

Thank you for submitting your manuscript to PLOS ONE. After careful consideration, we feel that it has merit but does not fully meet PLOS ONE’s publication criteria as it currently stands. Therefore, we invite you to submit a revised version of the manuscript that addresses the points raised during the review process.

Please make peer-to-peer modifications to the reviewer's comments.

We look forward to receiving your revised manuscript.

Kind regards,

Qian Wu

Academic Editor

PLOS ONE

Journal Requirements:

3. We note that your Data Availability Statement is currently as follows: [All relevant data are within the manuscript and its Supporting Information files.] Please confirm at this time whether or not your submission contains all raw data required to replicate the results of your study. Authors must share the “minimal data set” for their submission. PLOS defines the minimal data set to consist of the data required to replicate all study findings reported in the article, as well as related metadata and methods (https://journals.plos.org/plosone/s/data-availability#loc-minimal-data-set-definition). For example, authors should submit the following data: - The values behind the means, standard deviations and other measures reported; - The values used to build graphs; - The points extracted from images for analysis. Authors do not need to submit their entire data set if only a portion of the data was used in the reported study. If your submission does not contain these data, please either upload them as Supporting Information files or deposit them to a stable, public repository and provide us with the relevant URLs, DOIs, or accession numbers. For a list of recommended repositories, please see https://journals.plos.org/plosone/s/recommended-repositories. If there are ethical or legal restrictions on sharing a de-identified data set, please explain them in detail (e.g., data contain potentially sensitive information, data are owned by a third-party organization, etc.) and who has imposed them (e.g., an ethics committee). Please also provide contact information for a data access committee, ethics committee, or other institutional body to which data requests may be sent. If data are owned by a third party, please indicate how others may request data access.

Reviewers' comments:

Reviewer's Responses to Questions

**Comments to the Author**

1. Is the manuscript technically sound, and do the data support the conclusions?

Reviewer #1: Partly

Reviewer #2: Partly

Reviewer #3: Partly

2. Has the statistical analysis been performed appropriately and rigorously? 

Reviewer #1: Yes

Reviewer #2: I Don't Know

Reviewer #3: Yes

3. Have the authors made all data underlying the findings in their manuscript fully available?

Reviewer #1: No

Reviewer #2: No

Reviewer #3: Yes

4. Is the manuscript presented in an intelligible fashion and written in standard English?

Reviewer #1: Yes

Reviewer #2: Yes

Reviewer #3: No

5. Review Comments to the Author

Reviewer #1: This manuscript investigates the relationship between the Cardiometabolic Index (CMI) and gallstone disease (GSD) in U.S. adults. Using data from the National Health and Nutrition Examination Survey (NHANES) between 2017 and 2020, the study assesses whether CMI (a new composite indicator of obesity and lipid levels) associates non-linearly with GSD risk. The authors reveal that higher CMI correlates with an increased risk of GSD and propose that CMI might serve as a screening tool for GSD, with potential implications for clinical management.

In general, the manuscript is well-written and tables are clear. The authors use multiple ways (logistic regression, restricted cubic spline models, and subgroup analysis) to prove the positive correlation between CMI and GSD. The topic is beneficial to the clinical engagement and clearly suitable for publication in this journal. However, following issues need to be addressed before considering for publication.

1. This manuscript states they use logistics regression and restricted cubic spline models but haven’t provided any formulas or parameters deployed in this study. Need to address the rationale of choosing these two models for evaluating CMI – GSD correlation.

2. Lipid balance is crucial for gallstone disease, why choose CMI as indicator? Not [TG (mmol/L)/HDL-C (mmol/L)]? How does CMI compares to [TG (mmol/L)/HDL-C (mmol/L)] alone? And how does CMI compares to WHtR alone?

2. Lack visuals for logistics regression results and subgroup analysis.

3. Processed data (2825 out of 15,560) is not provided.

4. Although adjustments for multiple covariates were made, some potential confounders like diet habits, physical activity, and genetic predisposition are not accounted for. Discussing their potential impact on the results would improve the study's robustness.

5. Expand bias discussion (lack of other important factors as above). Address limitations of self-reported data -- could introduce recall bias. Highlight this as a limitation and consider suggesting longitudinal studies for validation.

6. Provide more insights on the subgroup analysis in the results, particularly regarding gender differences. Emphasize the clinical implications of gender-based variations in CMI's impact on GSD.

7. While the study indicates that CMI could be a screening tool, it may be helpful to discuss how CMI could complement existing GSD screening practices. Elaborate on how clinicians might integrate CMI in routine evaluations.

8. table 1: need to clearly indicate readers they are “logistic regression” models. The 4th “Q3” should be Q4.

9. figure 2 is too rough. It shows “figure 1” at the top left corner. Need more comprehensive captions explaining all elements including in the plot like the shade and cross-point. Enlarge x-axis and y-axis font size.

Reviewer #2: The present study demonstrates a nonlinear relationship between higher Cardiometabolic Index (CMI) and an increased risk of gallstone disease (GSD). However, there are several concerns regarding the methods used in the study that merit further consideration

Reviewer #3: This research is aligned with other research on cardiometabolic disease and liver disease such as GSD metabolic-associated fatty liver disease (MAFLD), applying different indices and statistical methods to test the association. The biological mechanism and underlying assumptions regarding the associations have been clearly defined previously, and possible co-variates/confounders have been identified, but the questions regarding which CM index is the most sensitive and specific clinical indicator for possible imminent liver disease in different populations residing in different environments and whether a valid universal threshold may be derived from the analyses if the association between the outcome and the index is not linear are not yet been answered. Although this study applied weighting in their analysis and adjusted for all relevant confounders in the multivariate logistic regression analyses, linearity was not observed. Restricted cubic spline analyses were thus performed to test for a ‘dose-dependent’ relationship, but failed to identify a threshold.

Some references that may be useful for the introduction, methodology or discourse are:

Global Epidemiology of Gallstones in the 21st Century: A Systematic Review and Meta-Analysis

Wang, Xin et al. Clinical Gastroenterology and Hepatology, Volume 22, Issue 8, 1586 – 1595

Zhang J, Liang D, Xu L, Liu Y, Jiang S, Han X, Wu H and Jiang Y (2024) Associations

between novel anthropometric indices and the prevalence of gallstones among

6,848 adults: a cross-sectional study. Front. Nutr. 11:1428488. doi: 10.3389/fnut.2024.1428488

Koyama, A. K., McKeever Bullard, K., Xu, F., Onufrak, S., Jackson, S. L., Saelee, R., Miyamoto, Y., & Pavkov, M. E. (2024). Prevalence of Cardiometabolic Diseases Among Racial and Ethnic Subgroups in Adults - Behavioral Risk Factor Surveillance System, United States, 2013-2021. MMWR. Morbidity and mortality weekly report, 73(3), 51–56. https://doi.org/10.15585/mmwr.mm7303a1

Duan S, Yang D, Xia H, Ren Z, Chen J, Yao S. Cardiometabolic index: A new predictor for metabolic associated fatty liver disease in Chinese adults. Front Endocrinol (Lausanne). 2022 Sep 16;13:1004855. doi: 10.3389/fendo.2022.1004855. PMID: 36187093; PMCID: PMC9523727.

Yan, L., Hu, X., Wu, S. et al. Association between the cardiometabolic index and NAFLD and fibrosis. Sci Rep 14, 13194 (2024). https://doi.org/10.1038/s41598-024-64034-3. https://rdcu.be/d0CD1

Song, Jimei, Yimei Li, Junxia Zhu, Jian Liang, Shan Xue, and Zhangzhi Zhu. "Non-linear Associations of Cardiometabolic Index with Insulin Resistance, Impaired Fasting Glucose, and Type 2 Diabetes among US Adults: A Cross-sectional Study." Frontiers in Endocrinology 15, (2024): 1341828. Accessed November 18, 2024. https://doi.org/10.3389/fendo.2024.1341828.

Lazzer, S., D'Alleva, M., Isola, M., De Martino, M., Caroli, D., Bondesan, A., Marra, A., & Sartorio, A. (2023). Cardiometabolic Index (CMI) and Visceral Adiposity Index (VAI) Highlight a Higher Risk of Metabolic Syndrome in Women with Severe Obesity. Journal of clinical medicine, 12(9), 3055. https://doi.org/10.3390/jcm12093055

DS Prasad, Zubair Kabir, JP Suganthy, AK Dash & BC Das. (‎2013)‎. Appropriate anthropometric indices to identify cardiometabolic risk in South Asians. WHO South-East Asia Journal of Public Health, 2 (‎3-4)‎, 142 - 148. World Health Organization. Regional Office for South-East Asia. https://iris.who.int/handle/10665/329790

Yan Zheng, Min Xu, Yanping Li, Adela Hruby, Eric B. Rimm, Frank B. Hu, Janine Wirth, Christine M. Albert, Kathryn M. Rexrode, JoAnn E. Manson, and Lu Qi. Gallstones and Risk of Coronary Heart Disease: Prospective Analysis of 270 000 Men and Women From 3 US Cohorts and Meta-Analysis

Arteriosclerosis, Thrombosis, and Vascular Biology Volume 36, Number 9 https://doi.org/10.1161/ATVBAHA.116.307507

Indices:

LAP

Ebrahimi, M., Seyedi, S. A., Nabipoorashrafi, S. A., Rabizadeh, S., Sarzaeim, M., Yadegar, A., Mohammadi, F., Bahri, R. A., Pakravan, P., Shafiekhani, P., Nakhjavani, M., & Esteghamati, A. (2023). Lipid accumulation product (LAP) index for the diagnosis of nonalcoholic fatty liver disease (NAFLD): a systematic review and meta-analysis. Lipids in health and disease, 22(1), 41. https://doi.org/10.1186/s12944-023-01802-6

Triglyceride-glucose index:

Li, H., Zhang, C. Association between triglyceride-glucose index and gallstones: a cross-sectional study. Sci Rep 14, 17778 (2024). https://doi.org/10.1038/s41598-024-68841-6

Aksoy E, Ergenç Z, Ocak Ök, Ergenç H. Triglyceride-Glucose Index is a Reliable Predictor of Metabolic Disorder in Gallstones. Bezmialem Science. 2024 Jul;12(3):363-367. doi:10.14235/bas.galenos.2024.43043

Body roundness index (BRI):

Wei C, Zhang G. Association between body roundness index (BRI) and gallstones: results of the 2017-2020 national health and nutrition examination survey (NHANES). BMC Gastroenterol. 2024 Jun 5;24(1):192. doi: 10.1186/s12876-024-03280-1. PMID: 38840060; PMCID: PMC11155175.

The threshold for risk assessment:

Eastwood, S. V., Hemani, G., Watkins, S. H., Scally, A., Davey Smith, G., & Chaturvedi, N. (2024). Ancestry, ethnicity, and race: explaining inequalities in cardiometabolic disease. Trends in molecular medicine, 30(6), 541–551. https://doi.org/10.1016/j.molmed.2024.04.002

General comments

The strength of the manuscript lies in the accommodation of relevant confounders and the application of restricted cubic spline analyses as a novel approach to make sense of the non-linearity of the association between this specific outcome and index.

Consider making more detailed and stronger deductions from the observed results, e.g., the subgroup analyses that were conducted to test for interactions between CMI and specific covariates. What stood out was the preponderance of males rather than females in the highest quartile (62.16 vs 37.8%) and the increasing trend in male proportions across quartiles with a reversed trend in females. This may be highlighted as a novel finding.

Specific comments:

There are numerous syntax errors throughout that need attention.

The Lipid Accumulation Product (LAP) index is mentioned in a sub-title within the Results section. This is the only occurrence of the term. Please clarify its relevance, because the paragraph reports on the CMI and the formulas for the two indices’ calculation differ.

The first paragraph in the introduction addresses Gallstone disease in general. The references are rather outdated. Please consider replacing them or adding more recent references (some of which are included in the above list of references that may be of interest to the authors)

In the third paragraph of this section, the CMI is introduced. You may consider referring to other more recently applied indices and state why you chose the CMI. E.g., that it has better predictive potential, as alluded to in the discussion section You refer to some of these indices in the paragraph within the discussion section that starts with ‘The CMI, as an indicator to assess the degree of obesity and lipid levels…’, but referring to them in the introduction section reflects insight within this field.

This is difficult, but consider re-writing the Conclusion so that it captures the contribution of this study to the body of scientific knowledge more distinctly.

Thank you for choosing this journal for your submission. Best wishes for your future research!

6. PLOS authors have the option to publish the peer review history of their article (what does this mean? ). If published, this will include your full peer review and any attached files.

**Do you want your identity to be public for this peer review?** For information about this choice, including consent withdrawal, please see our Privacy Policy .

Reviewer #1: No

Reviewer #2: No

Reviewer #3: **Yes: ** Rhena Delport

---

## [Author Response · Author response to Decision Letter 1]

7 Feb 2025

Dear Editors and Reviewers:

Thank you for your letter and for the reviewers’ comments concerning our manuscript entitled “Non-linear association of cardiometabolic index with gallstone disease in US adults: a cross-sectional study” (ID: 24-47105). Those comments are all valuable and very helpful for revising and improving our paper, as well as the important guiding significance to our research. We have studied the comments carefully and have made a correction. Revised portion are marked in red in the tracked version of manuscript. The final version of our manuscript has been submitted. The main corrections in the paper and the responses to the reviewer’s comments are as flowing:

Responds to the reviewer’s comments:

Reviewer #1:

1.This manuscript states they use logistics regression and restricted cubic spline models but haven’t provided any formulas or parameters deployed in this study. Need to address the rationale of choosing these two models for evaluating CMI – GSD correlation.

Response: We gratefully appreciate your valuable suggestion.We added the rationale of logistics regression and restricted cubic spline models to evaluate the correlation of CMI-GSD in the statistical analysis section of the manuscript.

2.Lipid balance is crucial for gallstone disease, why choose CMI as indicator? Not [TG (mmol/L)/HDL-C (mmol/L)]? How does CMI compares to [TG (mmol/L)/HDL-C (mmol/L)] alone? And how does CMI compares to WHtR alone?

Response: This is a very good suggestion. After synthesizing previous research findings, we posit that compared to individual lipid indices or waist-to-height ratio (WHtR), the cardiometabolic index (CMI) may provide a more comprehensive reflection of human metabolic status. It is noteworthy that we will conduct comparative analyses of these parameters in subsequent multicenter randomized controlled clinical trials to more directly demonstrate the advantages of the CMI index.

3.Lack visuals for logistics regression results and subgroup analysis.

Response: Thanks for your reminder. We have put the forest diagram into Figures 2 and 3.

4.Processed data (2825 out of 15,560) is not provided.

Response: We have uploaded the filtered data to https://figshare.com, DOI: 10.6084/m9.figshare.28323725.

5.Although adjustments for multiple covariates were made, some potential confounders like diet habits, physical activity, and genetic predisposition are not accounted for. Discussing their potential impact on the results would improve the study's robustness.

Response: We think this is an excellent suggestion.We have included physical activity in the covariate and adjusted it to improve the robustness of the results. However, due to the limitations of the NHANES database, diet habits and genetic predisposition are not currently available in the database.

6.Expand bias discussion (lack of other important factors as above). Address limitations of self-reported data -- could introduce recall bias. Highlight this as a limitation and consider suggesting longitudinal studies for validation.

Response: Thank you for the good advice. We have supplemented and highlighted the limitations of this study in the last paragraph of the discussion.

7.Provide more insights on the subgroup analysis in the results, particularly regarding gender differences. Emphasize the clinical implications of gender-based variations in CMI's impact on GSD.

Response: Thank you for your comments. We have added on lines 3 to 7 of page 10.

8.While the study indicates that CMI could be a screening tool, it may be helpful to discuss how CMI could complement existing GSD screening practices. Elaborate on how clinicians might integrate CMI in routine evaluations.

Response: We think this is an excellent suggestion. We have already added lines 250 to 255 in this manuscript.

9.table 1: need to clearly indicate readers they are “logistic regression” models. The 4th “Q3” should be Q4.

Response: Thank you for the good advice. We have supplemented the title in Table 1 and corrected "Q3" to "Q4" in Table 2.

10.figure 2 is too rough. It shows “figure 1” at the top left corner. Need more comprehensive captions explaining all elements including in the plot like the shade and cross-point. Enlarge x-axis and y-axis font size.

Response: Thank the reviewer for reading our paper carefully and giving the above positive comments. We have reworked Figure 4 and supplemented it in the first paragraph on page 8.

Reviewer #2

1.What influence does the use of NHANES data, which is cross-sectional in nature, have on how we interpret the relationship between CMI and gallstone disease (GSD)?

Response: This is a very good suggestion. We must admit that there are some inherent limitations to this study. Because this study is a cross-sectional study based on NHANES data, a causal relationship between CMI and GSD cannot be determined. However, in the follow-up study, our team will conduct a multicenter randomized controlled clinical trial to verify the relationship between the two.

2.Would incorporating other biomarkers or cardiometabolic indicators (for example, HOMA-IR or LDL cholesterol levels) into the CMI calculation alter its relationship with gallstone disease?

Response: We think this is an excellent suggestion. Some previous studies have found that HOMA-IR and LDL cholesterol levels are closely related to obesity and lipid metabolism, but there are no appropriate calculation methods and studies to incorporate these indicators into CMI calculation. In previous studies, HOMA-IR and LDL cholesterol levels have been found to be significantly associated with the risk of gallstones. Therefore, combined with our study results, we concurred that the inclusion of HOMA-IR and LDL cholesterol levels in CMI calculation would not change their relationship with gallstones, but this needs to be verified by multi-center randomized controlled clinical trials.

3.How does the use of waist-to-height ratio (WHtR) as a component of CMI compare to other measures of central obesity, like waist circumference or waist-to-hip ratio, in terms of both benefits and limitations?

Response: The waist-to-height ratio (WHtR) eliminates potential biases in the assessment of central obesity caused by height by standardizing waist circumference relative to height. Compared to waist circumference (WC) or waist-to-hip ratio (WHR) alone, WHtR provides a more equitable measure of obesity for populations with significant height variations, such as individuals from different ethnic groups. Studies have shown that WHtR exhibits higher sensitivity and specificity than WC and WHR in predicting the risks of metabolic syndrome, cardiovascular diseases, and diabetes. However, it is important to acknowledge the limitations of WHtR. First, for individuals with extremely tall or short stature, WHtR may underestimate or overestimate obesity risk, necessitating the use of additional indicators (e.g., BMI) for a comprehensive assessment. Second, WC and WHR remain the core metrics recommended by most clinical guidelines, and the clinical application of WHtR has yet to be widely adopted.

4.What is the reliability of using self-reported data from the NHANES questionnaire to diagnose gallstones, and what biases or limitations might arise from relying on self-reported health conditions in epidemiological studies?

Response: Thank you for your comments. In this study, the use of self-reported data from the NHANES questionnaire as the diagnostic criterion for gallstones was based on prior research. However, it is important to acknowledge that reliance on self-reported health conditions in epidemiological studies may introduce biases such as recall bias (e.g., participants may forget or misinterpret diagnostic details) and potential underdiagnosis or misdiagnosis.

5.What are the primary limitations of using a cross-sectional study design to infer causal relationships between CMI and GSD, and how could longitudinal or experimental research designs enhance the validity of these findings?

Response: Thank you for the good advice. First, in a cross-sectional study, both the exposure (CMI) and the outcome (gallstones) are measured simultaneously, making it impossible to determine whether the exposure precedes the outcome. Second, although we controlled for known confounding factors through multivariate adjustments, unmeasured confounders (such as genetic predisposition and dietary patterns) may still influence the results. Lastly, cross-sectional studies only provide a snapshot at a single point in time and cannot reflect the dynamic relationship between cardiometabolic indices and gallstones. In a longitudinal study design, long-term follow-up can help determine whether the exposure precedes the outcome. Additionally, repeated measurements can reveal the dynamic association between metabolic indicators and the risk of gallstones. In experimental study designs, randomized controlled trials, through random allocation, can minimize confounding bias and directly assess the impact of interventions (such as improving metabolic indicators) on gallstone risk. Furthermore, Mendelian randomization can also be used to enhance the reliability of causal inference.

6.While the study demonstrates a significant link between CMI and GSD, what potential biases (such as selection bias, information bias from self-reported diagnoses, or measurement error) should be considered, and what measures could be implemented in future studies to reduce their impact?

Response: We think this is an excellent suggestion. First, this study should consider selection bias: the general community population may not be representative of hospital-based cohorts. Second, information bias: self-reported diagnoses of gallstones may introduce recall bias, particularly since asymptomatic cases are likely to be underreported. Finally, measurement error: errors may arise from the measurement of waist-to-height ratio. In future research, we can track the temporal changes in the development of CMI and gallstones to clarify the temporal sequence and dynamic relationships. Additionally, combining evidence from observational studies, randomized controlled trials, and Mendelian randomization can enhance the robustness of causal inference.

7.The authors propose that CMI could serve as an effective screening tool for identifying individuals at high risk for GSD. How practical is it to implement CMI for this purpose in clinical practice, particularly considering its connection with other metabolic disorders like obesity and diabetes? What limitations might arise, such as issues with cost, accessibility, or applicability across diverse patient populations?

Response: We sincerely thank the editor and all reviewers for their valuable feedback that we have used to improve the quality of our manuscript. We have supplemented the clinical application of CMI in the second paragraph of page 9.

Reviewer #3

1.This research is aligned with other research on cardiometabolic disease and liver disease such as GSD metabolic-associated fatty liver disease (MAFLD), applying different indices and statistical methods to test the association. The biological mechanism and underlying assumptions regarding the associations have been clearly defined previously, and possible co-variates/confounders have been identified, but the questions regarding which CM index is the most sensitive and specific clinical indicator for possible imminent liver disease in different populations residing in different environments and whether a valid universal threshold may be derived from the analyses if the association between the outcome and the index is not linear are not yet been answered. Although this study applied weighting in their analysis and adjusted for all relevant confounders in the multivariate logistic regression analyses, linearity was not observed. Restricted cubic spline analyses were thus performed to test for a ‘dose-dependent’ relationship, but failed to identify a threshold.

Response: Thank you for your comments.We conducted a threshold effect analysis of the relationship between CMI and GSD and included the results in Table 3.

---

## [Decision Letter · Decision Letter 1]

PONE-D-24-47105R1Non-linear association of cardiometabolic index with gallstone disease in US adults: a cross-sectional studyPLOS ONE

Dear Dr. Li,

Thank you for submitting your manuscript to PLOS ONE. After careful consideration, we feel that it has merit but does not fully meet PLOS ONE’s publication criteria as it currently stands. Therefore, we invite you to submit a revised version of the manuscript that addresses the points raised during the review process.

Please make revisions based on the comments of the reviewers.

We look forward to receiving your revised manuscript.

Kind regards,

Qian Wu

Academic Editor

PLOS ONE

Journal Requirements:

Reviewers' comments:

Reviewer's Responses to Questions

**Comments to the Author**

1. If the authors have adequately addressed your comments raised in a previous round of review and you feel that this manuscript is now acceptable for publication, you may indicate that here to bypass the “Comments to the Author” section, enter your conflict of interest statement in the “Confidential to Editor” section, and submit your "Accept" recommendation.

Reviewer #1: (No Response)

Reviewer #2: All comments have been addressed

Reviewer #3: (No Response)

2. Is the manuscript technically sound, and do the data support the conclusions?

Reviewer #1: Partly

Reviewer #2: Yes

Reviewer #3: No

3. Has the statistical analysis been performed appropriately and rigorously? 

Reviewer #1: Yes

Reviewer #2: I Don't Know

Reviewer #3: Yes

4. Have the authors made all data underlying the findings in their manuscript fully available?

Reviewer #1: Yes

Reviewer #2: Yes

Reviewer #3: Yes

5. Is the manuscript presented in an intelligible fashion and written in standard English?

Reviewer #1: Yes

Reviewer #2: Yes

Reviewer #3: No

6. Review Comments to the Author

Reviewer #1: regarding my previous comment #2, I would compare CMI to the lipid indicator alone to make the choice of CMI technically sound, rather than you "posit" CMI may be a better one. This comparision would be an impoartant evidence to support the comprehensive reflection of CMI. Especially, from the CMI formula, we can see its a combination of the lipid indicator and waist-to-height ratio. They should be most direct comparision that were not supposed to skip.

2.Lipid balance is crucial for gallstone disease, why choose CMI as indicator? Not [TG

(mmol/L)/HDL-C (mmol/L)]? How does CMI compares to [TG (mmol/L)/HDL-C

(mmol/L)] alone? And how does CMI compares to WHtR alone?

"Response: This is a very good suggestion. After synthesizing previous research

findings, we posit that compared to individual lipid indices or waist-to-height ratio

(WHtR), the cardiometabolic index (CMI) may provide a more comprehensive reflection

of human metabolic status. It is noteworthy that we will conduct comparative analyses

of these parameters in subsequent multicenter randomized controlled clinical trials to

more directly demonstrate the advantages of the CMI index."

Reviewer #2: The Author has answered all the comments. They also incorporated the corrections in the manuscript

Reviewer #3: Dear Authors

The following comments have not been addressed in your rebuttal:

General comments

The strength of the manuscript lies in the accommodation of relevant confounders and the application of restricted cubic spline analyses as a novel approach to make sense of the non-linearity of the association between this specific outcome and index.

Consider making more detailed and stronger deductions from the observed results, e.g., the subgroup analyses that were conducted to test for interactions between CMI and specific covariates. What stood out was the preponderance of males rather than females in the highest quartile (62.16 vs 37.8%) and the increasing trend in male proportions across quartiles with a reversed trend in females. This may be highlighted as a novel finding.

Specific comments:

There are numerous syntax errors throughout that need attention.

The Lipid Accumulation Product (LAP) index is mentioned in a sub-title within the Results section. This is the only occurrence of the term. Please clarify its relevance, because the paragraph reports on the CMI and the formulas for the two indices’ calculation differ.

The first paragraph in the introduction addresses Gallstone disease in general. The references are rather outdated. Please consider replacing them or adding more recent references (some of which are included in the above list of references that may be of interest to the authors)

In the third paragraph of this section, the CMI is introduced. You may consider referring to other more recently applied indices and state why you chose the CMI. E.g., that it has better predictive potential, as alluded to in the discussion section You refer to some of these indices in the paragraph within the discussion section that starts with ‘The CMI, as an indicator to assess the degree of obesity and lipid levels…’, but referring to them in the introduction section reflects insight within this field.

Consider re-writing the Conclusion so that it captures the contribution of this study to the body of scientific knowledge more distinctly.

7. PLOS authors have the option to publish the peer review history of their article (what does this mean? ). If published, this will include your full peer review and any attached files.

**Do you want your identity to be public for this peer review?** For information about this choice, including consent withdrawal, please see our Privacy Policy .

Reviewer #1: No

Reviewer #2: No

Reviewer #3: No

---

## [Author Response · Author response to Decision Letter 2]

29 Mar 2025

Dear Editors and Reviewers:

Thank you for your letter and for the reviewers’ comments concerning our manuscript entitled “Non-linear association of cardiometabolic index with gallstone disease in US adults: a cross-sectional study” (ID: 24-47105). Those comments are all valuable and very helpful for revising and improving our paper, as well as the important guiding significance to our research. We have studied the comments carefully and have made a correction. Revised portions are marked in red in the tracked version of manuscript. The final version of our manuscript has been submitted. The main corrections in the paper and the responses to the reviewer’s comments are as flowing:

Responds to the reviewer’s comments:

Reviewer #1:

1. Lipid balance is crucial for gallstone disease, why choose CMI as indicator? Not [TG (mmol/L)/HDL-C (mmol/L)]? How does CMI compares to [TG (mmol/L)/HDL-C (mmol/L)] alone? And how does CMI compares to WHtR alone?

Response: We sincerely thank the editor and all reviewers for their valuable feedback that we have used to improve the quality of our manuscript. We used ROC analysis to compare CMI with WHtR and TG/HDL and added this analysis to the article.

Reviewer #3

1. Consider making more detailed and stronger deductions from the observed results, e.g., the subgroup analyses that were conducted to test for interactions between CMI and specific covariates. What stood out was the preponderance of males rather than females in the highest quartile (62.16 vs 37.8%) and the increasing trend in male proportions across quartiles with a reversed trend in females. This may be highlighted as a novel finding.

Response: We sincerely appreciate the insightful observation. We've added this novel finding to the results section.

2. There are numerous syntax errors throughout that need attention. The Lipid Accumulation Product (LAP) index is mentioned in a sub-title within the Results section. This is the only occurrence of the term. Please clarify its relevance, because the paragraph reports on the CMI and the formulas for the two indices’ calculation differ.

Response: Thanks for your reminder. That was a writing error, and we've changed it to CMI.

3. The first paragraph in the introduction addresses Gallstone disease in general. The references are rather outdated. Please consider replacing them or adding more recent references (some of which are included in the above list of references that may be of interest to the authors)

Response: We gratefully appreciate your valuable suggestion. We have added some of the literature from the above list of references to the first paragraph of the introduction.

4. In the third paragraph of this section, the CMI is introduced. You may consider referring to other more recently applied indices and state why you chose the CMI. E.g., that it has better predictive potential, as alluded to in the discussion section You refer to some of these indices in the paragraph within the discussion section that starts with ‘The CMI, as an indicator to assess the degree of obesity and lipid levels…’ but referring to them in the introduction section reflects insight within this field. Consider re-writing the Conclusion so that it captures the contribution of this study to the body of scientific knowledge more distinctly.

Response: Thank the reviewer for reading our paper carefully and giving the above positive comments. We have added these recommendations in the third paragraph of the introduction and rewritten the Conclusion.

---

## [Decision Letter · Decision Letter 2]

PONE-D-24-47105R2Non-linear association of cardiometabolic index with gallstone disease in US adults: a cross-sectional studyPLOS ONE

Dear Dr. Li,

Thank you for submitting your manuscript to PLOS ONE. After careful consideration, we feel that it has merit but does not fully meet PLOS ONE’s publication criteria as it currently stands. Therefore, we invite you to submit a revised version of the manuscript that addresses the points raised during the review process.

Please make peer-to-peer modifications to the reviewer's comments.

We look forward to receiving your revised manuscript.

Kind regards,

Qian Wu

Academic Editor

PLOS ONE

Journal Requirements:

Reviewers' comments:

Reviewer's Responses to Questions

**Comments to the Author**

1. If the authors have adequately addressed your comments raised in a previous round of review and you feel that this manuscript is now acceptable for publication, you may indicate that here to bypass the “Comments to the Author” section, enter your conflict of interest statement in the “Confidential to Editor” section, and submit your "Accept" recommendation.

Reviewer #1: All comments have been addressed

Reviewer #3: (No Response)

2. Is the manuscript technically sound, and do the data support the conclusions?

Reviewer #1: Yes

Reviewer #3: Yes

3. Has the statistical analysis been performed appropriately and rigorously? 

Reviewer #1: Yes

Reviewer #3: Yes

4. Have the authors made all data underlying the findings in their manuscript fully available?

Reviewer #1: Yes

Reviewer #3: Yes

5. Is the manuscript presented in an intelligible fashion and written in standard English?

Reviewer #1: Yes

Reviewer #3: Yes

6. Review Comments to the Author

Reviewer #1: All my concerns are well addressed. The authors have imporved the study with analysis details along with the critical corrections to the manuscript. I would recommend to publish this manuscript.

Reviewer #3: Thank you for submitting the revised manuscript.

Although all comments have been addressed suitably, there are still a few suggestions for consideration.

Specific comments:

“In contrast (line 98)to previous BRI and LAP indicators, CMI …” Please precede the acronyms with the full terms.

Line 276: Consider replacing ‘literatures’ with ‘reports’.

In Vancouver style, "P-value" should be written with a capital "P" and without a hyphen, as "P value".

General comment:

Line 309: Consider preceding the Conclusion with a summative statement regarding the clinical relevance of these findings, such as:

"However, the clinical relevance of identifying emergent GSD using the CMI is extended by acknowledging the significant relationship between GSD and CVD, probably due to common causal pathways, as established through meta-analysis and meta-regression." [ref. Hasan R, Allahbakhshi F, Shlyk AD, Allahbakhshi K. Gallstones as a predictor of elevated cardiovascular disease risk: A meta-analysis and meta-regression of over 7.4 million participants. PLoS One. 2025 Mar 19;20(3):e0314661. doi: 10.1371/journal.pone.0314661. PMID: 40106516; PMCID: PMC11922230.

Best wishes for your future research!

7. PLOS authors have the option to publish the peer review history of their article (what does this mean? ). If published, this will include your full peer review and any attached files.

**Do you want your identity to be public for this peer review?** For information about this choice, including consent withdrawal, please see our Privacy Policy .

Reviewer #1: No

Reviewer #3: No

---

## [Author Response · Author response to Decision Letter 3]

6 Jun 2025

Dear Editors and Reviewers:

Thank you for your letter and for the reviewers’ comments concerning our manuscript entitled “Non-linear association of cardiometabolic index with gallstone disease in US adults: a cross-sectional study” (ID: 24-47105). Those comments are all valuable and very helpful for revising and improving our paper, as well as the important guiding significance to our research. We have studied the comments carefully and have made a correction. Revised portions are marked in red in the tracked version of manuscript. The final version of our manuscript has been submitted. The main corrections in the paper and the responses to the reviewer’s comments are as flowing:

Responds to the reviewer’s comments:

Reviewer #3:

1. “In contrast (line 98) to previous BRI and LAP indicators, CMI …” Please precede the acronyms with the full terms.

Response: We sincerely thank the editor and all reviewers for their valuable feedback that we have used to improve the quality of our manuscript. We have added the full terms before the acronym

2. Line 276: Consider replacing ‘literatures’ with ‘reports’.

Response: Thanks for your reminder. We have made corrections

3. In Vancouver style, "P-value" should be written with a capital "P" and without a hyphen, as "P value".

Response: We sincerely appreciate the insightful observation. We change all the "P-value" in the manuscript to "P-value".

General comment:

4. Line 309: Consider preceding the Conclusion with a summative statement regarding the clinical relevance of these findings, such as: "However, the clinical relevance of identifying emergent GSD using the CMI is extended by acknowledging the significant relationship between GSD and CVD, probably due to common causal pathways, as established through meta-analysis and meta-regression."

Response: Thank the reviewer for reading our paper carefully and giving the above positive comments. We have added these recommendations in line 276.

---

## [Decision Letter · Decision Letter 3]

Non-linear association of cardiometabolic index with gallstone disease in US adults: a cross-sectional study

PONE-D-24-47105R3

Dear Dr. Li,

We’re pleased to inform you that your manuscript has been judged scientifically suitable for publication and will be formally accepted for publication once it meets all outstanding technical requirements.

Kind regards,

Qian Wu

Academic Editor

PLOS ONE

Additional Editor Comments (optional):

Reviewers' comments:

Reviewer's Responses to Questions

**Comments to the Author**

1. If the authors have adequately addressed your comments raised in a previous round of review and you feel that this manuscript is now acceptable for publication, you may indicate that here to bypass the “Comments to the Author” section, enter your conflict of interest statement in the “Confidential to Editor” section, and submit your "Accept" recommendation.

Reviewer #3: All comments have been addressed

2. Is the manuscript technically sound, and do the data support the conclusions?

Reviewer #3: Yes

3. Has the statistical analysis been performed appropriately and rigorously? 

Reviewer #3: Yes

4. Have the authors made all data underlying the findings in their manuscript fully available?

Reviewer #3: Yes

5. Is the manuscript presented in an intelligible fashion and written in standard English?

Reviewer #3: Yes

6. Review Comments to the Author

Reviewer #3: (No Response)

7. PLOS authors have the option to publish the peer review history of their article (what does this mean? ). If published, this will include your full peer review and any attached files.

**Do you want your identity to be public for this peer review?** For information about this choice, including consent withdrawal, please see our Privacy Policy .

Reviewer #3: **Yes: ** Rhena Delport
